# Venous leg ulcer healing as a determinant of quality of life in patients treated with unna boot: A quasi-experimental study

**Mário Lins Galvão de Oliveira**[1☉], **Felipe León-Morillas**[2‡], **Isadora Costa Andriola**[3☉], **Carmelo Sergio Gómez Martínez**[4‡], **Bruno Araújo da Silva Dantas**[5‡*], **Gilson de Vasconcelos Torres**[6‡]

1 Centro de Ciências da Saúde, Universidade Federal do Rio Grande do Norte, Natal, RN, Brazil, 2 Departamento de Fisioterapia, Universidad Católica de Murcia (UCAM), Murcia, Spain, 3 Departamento de Atenção à Saúde, Universidade Federal do Rio Grande do Norte, Natal, RN, Brazil, 4 Facultad de Enfermería, Universidad Católica de Murcia (UCAM), Murcia, Spain, 5 Faculdade de Ciências da Saúde do Trairi, Universidade Federal do Rio Grande do Norte, Santa Cruz, RN, Brazil, 6 CNPQ (Researcher PQ1D), Centro de Ciências da Saúde, Universidade Federal do Rio Grande do Norte, Natal, RN, Brazil

☉ These authors contributed equally to this work.
‡ FL-M, CSGM, BASD and GVT also contributed equally to this work.
* bruno.dantas@ufrn.br

**Data Availability Statement:** The dataset and analyses were deposited on the Mendeley Data platform, available through the doi: 10.17632/

## Abstract

Our objective was to assess the effect of Venous Leg Ulcer (VLU) healing on Quality of Life (QoL) in patients undergoing compression therapy. This non-randomized, quasi-experimental, and observational study involved patients with VLU. A convenience sample of individuals receiving services was followed for at least one year while undergoing compression therapy. The Medical Outcomes Short-Form Health Survey (SF-36) and the Charing Cross Venous Ulcer Questionnaire (CCVUQ) were employed to measure the variables of interest. Study participants were categorized into the Healed Group (HG) and the Unhealed Group (UG). The final sample comprised 103 individuals. The HG demonstrated improvements in SF-36 scores in the domains of Social Role Functioning (n = 34, p<0.001), Physical Role Functioning (n = 33, p<0.001), and the Physical Health Dimension (n = 38, p<0.001). Additionally, the CCVUQ assessment revealed score enhancements in the domains of Domestic Activities (n = 30, p = 0.001) and Social Interaction (n = 30, p = 0.009). QoL showed significant improvements in functionality, physical performance, and social interaction in the HG after one year of compression therapy treatment. In contrast, the UG was the only group to experience significant deteriorations in QoL.

## Introduction

Venous Leg Ulcers (VLU) represent a significant global health challenge, recognized as a consequence of the pathological impacts of chronic venous insufficiency [1]. They have a prevalence of 0.32% and an incidence of 0.17% in the adult population, predominantly affecting women and older adults [2]. The treatment of VLUs is often costly, posing a substantial concern for low-income populations dependent on public healthcare systems [3]. The debilitating

3vjd66g9gc.1. URL: https://data.mendeley.com/datasets/3vjd66g9gc/1.

**Funding:** This study was funded by the National Council for Scientific and Technological Development (Brazil) through the CNPq/MCTI/FNDCT call, grant number 408535/2021-0 and document n° 18/2021 - Tier B, for consolidated groups. The grant was awarded to researcher Dr. Gilson de Vasconcelos Torres, from the Federal University of Rio Grande do Norte, Brazil (Level PQ1D Researcher). The funders had no role in study design, data collection and analysis, decision to publish, or preparation of the manuscript.

**Competing interests:** The authors have declared that no competing interests exist.

nature of VLUs affects various aspects of life, including emotional, social, and physical well-being-all crucial components of human Quality of Life (QoL) [4].

Generally, QoL is understood as an individual's perception of their placement within their cultural, social, and spiritual contexts, encompassing their beliefs, perspectives, satisfaction, and achievements [5]. For individuals with VLUs, QoL may also include specific aspects related to the ulcer, such as concerns about body cosmesis and the physical limitations imposed by the condition [6]. In this light, the occurrence of VLU, compounded bysocial, economic, and pathological factors, increases the likelihood of prolonged treatments and recurrence rates. One study has shown that healing time for this demographic is extended in 93.0% of cases, requiring up to 12 months for complete healing, while 7.0% remain unhealed after five years [4]. Therefore, when devising strategies to address the needs of this population, it is imperative to incorporates QoL assessments, not merely focusing on clinical outcomes. This approach requires that patients understand the influence of their thoughts and attitudes on their health outcomes [7].

Most existing research on VLUs has focused on healing or treatment outcomes, neglecting other significant patient-centered variables, such as improvements in QoL and overall well-being [8]. A study 2022 study conducted in India found that physical limitations and the inability to perform daily life activities were associated with lower QoL scores [9]. Another study from Brazil highlighted the impact of pain on various QoL aspects, emphasizing its debilitating effect [10]. However, few studies have evaluated or quantified improvements in these aspects through the application of available treatments and adjunct technologies.

To bridge this gap, our research aimed to assess the effect of VLU healing on QoL in patients undergoing compression therapy. We hypothesized that individuals with healed VLUs would exhibit more significant improvements in QoL than those who did not heal after one year of treatment.

## Materials and methods

### Ethical aspects

In accordance with Brazilian legislation for conducting research involving human subjects and aligned with the Helsinki Declaration on scientific best practices, this study was approved by the Research Ethics Committee of the Federal University of Rio Grande do Norte, under opinion number 2.322.176. The involved institutions provided prior authorization for the research. Before applying any instruments, participants were informed of the study's purpose, risks, and benefits, and signed an informed consent form to participate. Participants' consent was provided through a signed written informed consent form. Neither participants nor researchers received any form of remuneration or reward.

### Study design and location

This was a non-randomized, quasi-experimental, and observational study with a quantitative approach involving patients with VLUs who were followed at a specialized chronic wound treatment service. This service was affiliated with 29 Primary Health Care (PHC) units in Parnamirim, Rio Grande do Norte, Brazil. The research was conducted from August 2020 to November 2021.

### Population and sample

The target population consisted of individuals with active chronic VLU treated in PHC units. An active VLU was defined as one that had remained open for at least three months without

healing. A convenience sample was selected. Initial mapping of local PHC units identified 157 patients with VLUs in the study area. The sample size was calculated to be 112 participants, based on a 95% confidence level and a 5% margin of error, using an online calculator for finite populations. However, a total of 103 individuals completed the study. The sample size calculation was performed using a formula for finite populations via an online calculator: https://calculareconverter.com.br/calculo-amostral/.

Inclusion criteria included individuals 18 years or older, registered at any PHC unit covered by the service, and presenting with at least one active VLU (Unhealed VLU). Exclusion criteria included those with completely healed VLUs before the study's start, those who missed appointments for more than a month, those who passed away or moved out of the coverage area after initial data collection, and patients with leg ulcers of mixed or non-venous origin. The theoretical framework for this study adopted the concept that mixed or non-venous leg ulcers include arterial ulcers or those with both venous and arterial involvement [11]. The wound etiology was determined with assistance from the Clinical-Etiology-Anatomy-Pathophysiology (CEAP) classification system [12]. Patients not already receiving specialized service from PHC units were also included through identification and recruitment by PHC professionals.

## Instruments and variables

A structured form was used to collect sociodemographic, health, clinical, and care data. This form, developed by the researchers, included closed-ended questions divided into response categories, assessing aspects such as gender, marital status, occupation, housing situation, age group, education level, and income (with the minimum wage in Brazil in 2021 set at BRL 1,100.00, approximately USD 197.00). Variables related to Health Status and Habits were the presence/absence of chronic diseases, alcohol and tobacco use, allergies, continuous medication use, changes in bladder and bowel elimination, personal hygiene, mobility, and mental state.

To measure QoL, the Portuguese version of the Medical Outcomes Short-Form Health Survey (SF-36) was used, consisting of 36 Likert-scale questions assessing eight domains and two dimensions, including physical, functional, emotional, mental health, and self-perception of general health. Score ranged from 0 to 100, with lower scores indicating worse QoL [5]. The Charing Cross Venous Ulcer Questionnaire (CCVUQ), a specific QoL assessment tool for individuals with VLU, was also used. It consists of 21 questions evaluating four divisions: Domestic Activities, Social Interaction, Cosmesis, and Emotional State. Scores ranged from 0 to 100, with lower scores indicating better QoL [13]. The two instruments were chosen for their complementary nature, providing a broader assessment of QoL.

The sample was divided into two groups based on their outcomes after a minimum interval of one year (T1-T2): the Unhealed Group (UG), comprising those still undergoing VLU treatment, and the Healed Group (HG), consisting of those whose VLUs had healed. All study variables were analyzed following this division.

## Follow-up consultations

Follow-up consultations were scheduled, at most, every seven days. During these appointment, a nurse changed the primary dressing of the VLU, and the lesion's progress was evaluated with a general practitioner. In some more complex cases, dressing changes were scheduled twice weekly such as when there was a large volume of exudate. Between the two data collection points, the general practitioner implemented treatments based on current literature. The primary treatment was compressive therapy using an Unna boot, a 10.2 cm-wide and 9.14 m-

long bandage that can be cut to the desired size. The compression level of compression, determined by a medical professional, was indicated visually by the bandage. The bandage composition includes zinc oxide, glycerin, distilled water, and gelatin [14]. At each dressing change appointment at the service, a new evaluation was conducted by the healthcare professional. The evaluation focused on the same aspects, including the size and width of the wound, the appearance of its edges and interior, as well as the presence and characteristics of exudate. Patients were routinely instructed on how to care for their dressings at home, such as protecting them during bathing to avoid getting them wet, following a healthy diet, and elevating the leg above heart level.

A vascular specialist performed an initial a comprehensive evaluation each patient, including measurement of the Ankle-Brachial Index, which is a validated method for assessing obstructive disease. This measurement ruled out obstructive arterial disease and ensured safe use of compression therapy, which is contraindicated for ulcers with arterial or mixed etiology [14, 15]. Any contraindications to the use of Unna boot therapy, such as infection, were addressed before starting treatment. Although other therapies were available, the Unna boot was the most advanced technology offered at the service. Patients were instructed to return to the service if they experienced symptoms such as increased pain, excessive leakage, bleeding, or other complications. A vascular surgeon was available for special cases, but did not oversee all patients directly. Patients or caregivers applied secondary dressings at home or in other facilities. To be considered healed, the VLU area should be completely covered by epithelial tissue, with no evidence of granulation. This assessment was conducted by the physician in the service.

## Data collection and availability

Data were collected through face-to-face interviews at two points in time. The first data collection (T1) occurred between August and October 2020, and the second (T2) between September and November 2021. The interviews were conducted by a trained team of master's and doctoral students in health sciences, a doctor, and two nurses. Instruments were applied during routine dressing changes, with a minimum of 12 months between interviews. Care was taken to collect data before opening the dressing to minimize emotional bias. No intervention or recommendation, were provided by researchers. Data were deposited on the Mendeley Data platform, available through the doi: 10.17632/3vjd66g9gc.1 and URL: https://data.mendeley.com/datasets/3vjd66g9gc/1.

## Data analysis and processing

Data were tabulated and organized using Microsoft Excel (Microsoft Corporation, Washington, WA, USA) and presented in tables. They were subsequently exported for statistical analysis using the Statistical Package for Social Science for Windows (SPSS) version 20.0 (IBM, Armonk, NY, USA). The Kolmogorov-Smirnov test revealed that the sample was non-normal.

Internal consistency of the QoL scales (SF-36 and CCVUQ) was evaluated using Cronbach's Alpha, with the following thresholds: $0.30 < \alpha \leq 0.60$ (low); $0.60 < \alpha \leq 0.75$ (moderate); $0.75 < \alpha \leq 0.90$ (high); $\alpha > 0.90$ (very high) [16]. The SF-36 showed moderate consistency ($\alpha = 0.68$) while the CCVUQ had high consistency ($\alpha = 0.89$). A descriptive analysis was conducted, which included absolute and relative frequencies of categorical variables, such as the sociodemographic and clinical profile of individuals with VLUs. Additionally, means, Standard Deviations (SD), and percentiles (25th, 50th, and 75th) of scalar QoL scores were calculated. Differences between means were assessed using Pearson's Chi-Square Test or Fisher's Exact Test, as appropriate, and the Mann-Whitney U test was used to assess

differences between domains. The Sign test was applied to compare QoL between two time points. Logistic Regression was performed on variables of interest, and results were synthesized using $R^2$ (Cox & Snell), the value of the LR Model, Hosmer Lemeshow Test, Beta (ß), Exp (ß). A 5% margin of error and a 95% Confidence Interval (CI) were adopted, with a significance set at $p < 0.05$ [17].

## Results

The initial sample of the study consisted of 157 patients. Of these, 45 could not be located at addresses provided by the PHC units. Therefore, 112 participants were included at the first data collection point (T1). However, 6 were excluded due to death, and 3 due to relocation outside the service coverage area. A final sample of n = 103 individuals was obtained, with n = 43 (41.7%) progressing to healing after one year of treatment (HG). The recruitment and selection process of the sample is depicted in Fig 1.

In Table 1, the sociodemographic and health profiles of the groups appear similar. Among the sociodemographic characteristics, the total sample showed a predominance of the female gender (n = 75/72.8%/p = 0.227), individuals aged 60 years or older (n = 68/66.0%/p = 0.128), those with an income up to one minimum wage (n = 85/82.5%/p = 0.600), low educational attainment (n = 82/79.6%/p = 0.908), and individuals with a profession or occupation (n = 82/79.6%/p = 0.908). Among health characteristics, daily medication use was notably prevalent (n = 81/78.6%/p = 0.691). Other variables potentially favoring VLU healing, such as the absence of smoking or alcoholism (n = 94/91.3%/p = 0.592), absence of allergies (n = 83/80.6%/p = 0.495), mobility impairment (n = 96/93.2%/p = 0.951), and mental state alterations (n = 99/96.1%/p = 0.733) were proportionally distributed between both groups (UG and HG).

Table 2 presents the QoL scores of the total sample at both data collection points (T1 and T2). In the SF-36 assessment conducted, there was an increase in the average scores in the General health perceptions (p<0.001), Social role functioning (p<0.001), Pain (p<0.001), Physical role functioning (p<0.001), Physical functioning (p<0.001) domains, and Physical health dimension. Meanwhile, a decrease in the average score was observed in the Emotional Role Domain (p<0.001). The CCVUQ assessment, although showing a trend towards increased averages, did not demonstrate statistical significance.

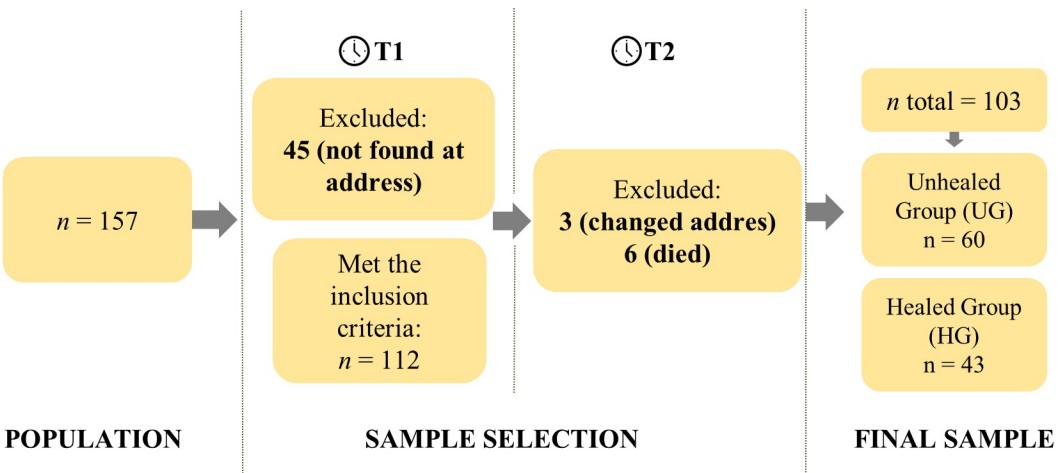

**Fig 1. Detailed flowchart of the recruitment and sample selection process and formation of study groups.**

**Table 1. Sociodemographic and health profile according to the outcome of the wound (T2).**

| Sociodemographic and Health Profile | Outcome of the VLU | | | | | Total (n = 103) | |
|---|---|---|---|---|---|---|---|
| | UG (n = 60) | | HG (n = 43) | | p [a] | | |
| | n | % | n | % | | n | % |
| **Sociodemographic Aspects** | | | | | | | |
| **Gender** | | | | | | | |
| Woman | 41 | 68.3 | 34 | 79.1 | 0.227 | 75 | 72.8 |
| Man | 19 | 31.7 | 9 | 20.9 | | 28 | 27.2 |
| **Age group** | | | | | | | |
| < 60 years | 24 | 40.0 | 11 | 25.6 | 0.128 | 35 | 34.0 |
| ≥ 60 years | 36 | 60.0 | 32 | 74.4 | | 68 | 66.0 |
| **Marital status** | | | | | | | |
| No companion | 29 | 48.3 | 24 | 55.8 | 0.454 | 53 | 51.5 |
| With partner | 31 | 51.7 | 19 | 44.2 | | 50 | 48.5 |
| **Housing Situation** | | | | | | | |
| Rented | 7 | 11.7 | 5 | 11.6 | 0. 977 | 12 | 11.7 |
| Own | 50 | 83.3 | 36 | 83.7 | | 86 | 83.5 |
| **Income** | | | | | | | |
| < 1 minimum wage | 51 | 85.0 | 34 | 79.1 | 0.600 | 85 | 82.5 |
| > 1 minimum wage | 7 | 11.7 | 8 | 18.6 | | 15 | 14.6 |
| **Education level [b]** | | | | | | | |
| Low | 48 | 80.0 | 34 | 79.1 | 0.908 | 82 | 79.6 |
| Hight | 12 | 20.0 | 9 | 20.9 | | 21 | 20.4 |
| **Profession / Occupation** | | | | | | | |
| No | 12 | 20.0 | 9 | 20.9 | 0.908 | 21 | 20.4 |
| Yes | 48 | 80.0 | 34 | 79.1 | | 82 | 79.6 |
| **Health aspects** | | | | | | | |
| **Systemic Arterial Hypertension** | | | | | | | |
| Yes | 38 | 63.3 | 24 | 55.8 | 0.442 | 62 | 60.2 |
| No | 22 | 36.7 | 19 | 44.2 | | 41 | 39.8 |
| **Diabetes Mellitus** | | | | | | | |
| Yes | 22 | 36.7 | 11 | 25.6 | 0.234 | 33 | 32.0 |
| No | 38 | 63.3 | 32 | 74.4 | | 70 | 68.0 |
| **Daily use of medications** | | | | | | | |
| Yes | 48 | 80.0 | 33 | 76.7 | 0.691 | 81 | 78.6 |
| No | 12 | 20.0 | 10 | 23.3 | | 22 | 21.4 |
| **Alcohol consumption/Smoking** | | | | | | | |
| Yes | 6 | 10.0 | 3 | 7.0 | 0.592 [c] | 9 | 8.7 |
| No | 54 | 90.0 | 40 | 93.0 | | 94 | 91.3 |
| **Allergies** | | | | | | | |
| Yes | 13 | 21.7 | 7 | 16.3 | 0.495 | 20 | 19.4 |
| No | 47 | 78.3 | 36 | 83.7 | | 83 | 80.6 |
| **Impaired mobility** | | | | | | | |
| Yes | 4 | 6.7 | 3 | 7.0 | 0.951 [c] | 7 | 6.8 |
| No | 56 | 93.3 | 40 | 93.0 | | 96 | 93.2 |
| **Impaired mental state** | | | | | | | |

(*Continued*)

**Table 1.** (Continued)

| Sociodemographic and Health Profile | Outcome of the VLU | | | | | Total (n = 103) | |
|---|---|---|---|---|---|---|---|
| | UG (n = 60) | | HG (n = 43) | | p ᵃ | | |
| | n | % | n | % | | n | % |
| Yes | 2 | 3.3 | 2 | 4.7 | 0.733 ᶜ | 4 | 3.9 |
| No | 58 | 96.7 | 41 | 95.3 | | 99 | 96.1 |

ᵃ: Pearson's Chi-Square Test.

ᵇ Education Level: Low: high school or less; Hight: higher education or higher.

ᶜ: Fisher's Exact Test.

Minimum Wage: in Brazil, the value was BRL 1,100 in the year 2021, approximately USD 197.

$\geq$: "greater than or equal to".

Table 3 highlights the difference in QoL scores between T1 and T2 across groups. Notably, there was a positive change in SF-36 scores in the HG for Physical role functioning (p<0.001), Physical functioning (p = 0.014), and Physical health Dimension (p = 0.046) domains, while the UG showed negative changes in these aspects. The Pain Domain exhibited an inverse trend, with a negative change in the HG and a positive one in the UG (p = 0.022). Conversely, the CCVUQ scores showed a significant negative variation in all aspects in the HG, while all aspects in the UG displayed positive changes (p<0.05).

A preliminary analysis comparing the groups at T1 revealed, that the UG had higher SF-36 scores at that time in the Physical role functioning domain (mean = 12.2, SD = 21.5) compared

**Table 2. Quality of life measured by SF-36 and CCVUQ (T1 and T2) of the total sample (n = 103).**

| QoL | T1 (n = 103) | | T2 (n = 103) | | p ᵃ |
|---|---|---|---|---|---|
| | Percentiles | Mean (SD) | Percentiles | Mean (SD) | |
| | 25–50–75 | | 25–50–75 | | |
| **SF-36—Domains** | | | | | |
| General health perceptions | 45.0–55.0–60.0 | 51.6 (14.2) | 50.0–70.0–75.0 | 63.8 (17.2) | <0.001 |
| Social role functioning | 12.5–37.5–37.5 | 30.0 (21.9) | 50.0–50.0–62.5 | 51.0 (10.4) | <0.001 |
| Emotional role functioning | 33.3–66.7–66.7 | 48.5 (29.8) | 0.0–0.0–66.7 | 28.2 (43.5) | <0.001 |
| Pain | 10.0–20.0–40.0 | 24.9 (19.4) | 20.0–30.0–60.0 | 37.8 (25.6) | <0.001 |
| Physical role functioning | 0.0–0.0–5.0 | 8.1 (17.8) | 0.0–20.0–55.0 | 31.7 (33.0) | <0.001 |
| Physical functioning | 0.0–0.0–0.0 | 8.2 (27.0) | 0.0–0.0–75.0 | 28.9 (43.5) | <0.001 |
| Mental health | 52.0–56.0–64.0 | 57.4 (8.2) | 48.0–60.0–64.0 | 56.1 (10.5) | 0.743 |
| Vitality | 45.0–50.0–60.0 | 52.1 (11.9) | 50.0–50.0–55.0 | 53.7 (9.5) | 0.445 |
| Total Score | 28.0–33.6–40.0 | 35.2 (9.2) | 35.2–38.4–57.4 | 43.9 (12.4) | <0.001 |
| **SF-36—Dimensions** | | | | | |
| Physical health | 23.0–27.0–33.0 | 29.0 (9.2) | 0.0–0.0–66.7 | 43.2 (11.1) | <0.001 |
| Mental health | 40.8–48.0–52.9 | 48.1 (9.6) | 48.0–60.0–64.0 | 50.5 (10.0) | 0.844 |
| **CCVUQ—Divisions** | | | | | |
| Cosmesis | 33.4–48.6–63.6 | 48.6 (18.6) | 20.7–54.7–80.5 | 53.0 (30.2) | 0.315 |
| Emotional status | 37.9–57.5–70.4 | 54.4 (20.1) | 20.7–56.8–78.7 | 53.6 (29.0) | 0.920 |
| Social interaction | 43.7–65.4–71.8 | 58.5 (20.1) | 21.6–51.7–75.7 | 49.9 (26.3) | 0.484 |
| Domestic activities | 49.9–71.3–83.2 | 63.5 (20.4) | 19.8–52.5–84.2 | 50.3 (29.2) | 0.056 |
| Total Score | 44.6–56.0–66.7 | 54.5 (15.6) | 21.0–52.5–77.0 | 51.3 (26.7) | 0.921 |

SD: Standard Deviation

ᵃ: Mann-Whitney U Test.

**Table 3. Variation in quality of life scores (SF-36 and CCVUQ) between T1 and T2 according to the study groups.**

| QoL | Outcomes of VLU (n = 103) | | | | |
|---|---|---|---|---|---|
| | UG (n = 60) | | HG (n = 43) | | $p^a$ |
| | Percentiles (T1-T2) | Mean (SD) | Percentiles (T1-T2) | Mean (SD) | |
| | 25–50–75 | | 25–50–75 | | |
| **SF-36—Domains** | | | | | |
| Physical role functioning | (-3.7)– 5.0–32.5 | 12.9 (33.4) | 5.0–40.0–65.0 | 38.5 (33.2) | <0.001 |
| Physical functioning | 0.0–0.0–0.0 | 11.2 (49.9) | 0.0–0.0–100.0 | 33.7 (45.6) | 0.014 |
| Pain | 0.0–20.0–40.0 | 19.0 (29.4) | (-20.0)– 10.0–30.0 | 4.2 (34.0) | 0.022 |
| Social role functioning | 3.1–25.0–37.5 | 21.5 (24.5) | 12.5–12.5–37.5 | 20.3 (23.0) | 0.820 |
| Vitality | (-1.0)– 0.0–10.0 | -0.1 (14.7) | (-5.0)– 5.0–10.0 | 3.9 (12.9) | 0.118 |
| Mental health | (-1.2)– 0.0–7.0 | -2.5 (12,4) | (-4.0)– 0.0–8.0 | 0.5 (14.3) | 0.410 |
| Emotional role functioning | (-66.7)–(-33.3)– 0.0 | -26.7 (54.8) | (-66.7)– 0.0–33.3 | -11.6 (57.7) | 0.180 |
| General health perceptions | 0.0–20.0–30.0 | 14.9 (21.6) | (-5.0)– 5.0–25.0 | 8.5 (20.6) | 0.139 |
| Total Score | (-3.1)– 2.8–13.2 | 6.1 (16.9) | 1.3–8.3–29.4 | 12.3 (15.7) | 0.041 |
| **SF-36—Dimensions** | | | | | |
| Physical health | 3.0–11.0–19.7 | 11.6 (15.4) | 7.0–15.0–29.0 | 17.8 (13.9) | 0.046 |
| Mental health | (-9.0)– 1.4–9.4 | 1.1 (15.6) | (-6.6)– 2.8–16.1 | 4.3 (15.6) | 0.234 |
| **CCVUQ—Divisions** | | | | | |
| Cosmesis | (-11.1)– 21.6–34.7 | 12.4 (31.6) | (-26.4)–(-6.4)– 19.0 | -6.7 (31.2) | 0.004 |
| Emotional status | (-13.5)– 9.2–25.5 | 5.0 (31.3) | (-36.4)–(-12.8)– 9.4 | -8.9 (30.7) | 0.022 |
| Social interaction | (-9.1)– 3.9–17.8 | 2.1 (28.3) | (-51.8)–(-18.5)– 1.0 | -23.5 (29.7) | <0.001 |
| Domestic activities | (-19.9)– 0.0–12.9 | -3.7 (30.0) | (-51.5)–(-35.4)– 0.0 | -26,3 (33,3) | 0.001 |
| Total Score | (-12.2)– 10.4–24.3 | 5.0 (26.1) | (-39.1)–(-19.3)– 7.0 | -14.7 (27.2) | 0.001 |

SD: Standard Deviation.

[a] Mann-Whitney U test.

to the HG (mean = 2.4, SD = 7.8, p = 0.004), and in the Physical Health Dimension (UG: mean = 31.0, SD = 10.5; HG: mean = 26.2, SD = 5.9, p = 0.021). In the CCVUQ assessment, the Social Interaction division showed higher scores in the UG (mean = 54.3, SD = 20.8) compared to the HG (mean = 64.4, SD = 17.6, p = 0.022). No significant differences were found in the other aspects evaluated.

To quantify the participants who improved in QoL scores between T1 and T2, Table 4 shows that the UG exhibited increases in SF-36 scores for most individuals in the Social role functioning (n = 45/ p<0.001), Physical role functioning (n = 33/ p = 0.014), General health perceptions (n = 40/ p = 0.001), and Pain (n = 40/ p = 0.001) domains. An reduction was noted in Emotional role functioning Domain (n = 40/ p<0.001) and Physical health Dimension (n = 50/ p<0.001). The UG also exhibited an increase in CCVUQ scores in the Cosmesis aspect (n = 40/ p = 0.009). In contrast, the HG showed increases in SF-36 scores for Social role functioning (n = 34/ p<0.001), Physical role functioning (n = 33/ p<0.001), and Physical health Dimension (n = 38/ p<0.001), with no significant change (=) in the Physical health Dimension (n = 24/ p<0.001). In the CCVUQ assessment, the HG showed reductions in its scores in the Domestic activities (n = 30/ p = 0.001) and Social interaction (n = 30/ p = 0.009) divisions. Importantly, the HG did not exhibit any significant worsening in QoL scores on either scale.

Additionally, logistic regression analysis of the most prominent aspects in the HG (Table 5) revealed that the SF-36 indicated greater significance in the Physical Role Functioning domain ($R^2$ = 0.124/ p<0.001/ ß = 0.022/ Exp (ß) = 1.01), while the CCVUQ demonstrated significance in the Social interaction division ($R^2$ = 0.157/ p<0.001/ ß = -0.029/ Exp (ß) = 0.97).

**Table 4. Evolution of quality of life (SF-36 and CCVUQ) between T1 and T2 according to the study groups.**

| QoL | Outcomes of VLU (T1-T2) (n = 103) | | | | | | | |
|---|---|---|---|---|---|---|---|---|
| | UG (n = 60) | | | | HG (n = 43) | | | |
| | + | = | - | p [a] | + | = | - | p |
| **SF-36—Domains** | | | | | | | | |
| Social role functioning | 45 | 8 | 7 | <0.001 | 34 | 4 | 5 | <0.001 |
| Physical role functioning | 33 | 12 | 15 | 0. 014 | 33 | 10 | 0 | <0.001 |
| General health perceptions | 40 | 6 | 14 | 0. 001 | 24 | 7 | 12 | 0. 067 |
| Pain | 40 | 6 | 14 | 0. 001 | 22 | 3 | 18 | 0. 635 |
| Physical functioning | 12 | 42 | 6 | 0. 014 | 18 | 24 | 1 | <0.001 |
| Emotional role functioning | 11 | 9 | 40 | <0.001 | 14 | 8 | 21 | 0. 310 |
| Mental health | 26 | 11 | 23 | 0. 775 | 18 | 8 | 17 | 1.000 |
| Vitality | 23 | 10 | 27 | 0. 671 | 23 | 9 | 11 | 0.059 |
| Total Score | 38 | 0 | 22 | 0. 053 | 34 | 0 | 9 | <0.001 |
| **SF-36—Dimensions** | | | | | | | | |
| Physical health | 10 | 0 | 50 | <0.001 | 38 | 0 | 5 | <0.001 |
| Mental health | 0 | 0 | 60 | 0. 519 | 18 | 8 | 17 | 0.222 |
| **CCVUQ—Divisions** | | | | | | | | |
| Domestic activities | 26 | 10 | 24 | 0.888 | 9 | 4 | 30 | 0. 001 |
| Cosmesis | 40 | 1 | 19 | 0.009 | 15 | 3 | 25 | 0. 155 |
| Social interaction | 34 | 2 | 24 | 0.237 | 12 | 1 | 30 | 0. 009 |
| Emotional status | 34 | 1 | 25 | 0.298 | 14 | 4 | 25 | 0. 109 |
| Total Score | 38 | 1 | 21 | 0. 037 | 14 | 0 | 29 | 0. 033 |

+: Increased score

=: Maintained score

-: Reduced score

[a]: Sign Test.

## Discussion

Our research data indicated improvements in QoL aspects between T1 and T2 data collection points in both study groups. However, the HG demonstrated significant improvements across a broader spectrum of aspects, as measured by both the SF-36 and CCVUQ instruments. Notably, enhancements in functionality, physical condition, and social aspects were most

**Table 5. Logistic regression analysis of the most significant aspects in the HG (n = 43).**

| QoL (HG) | $R^2$ (Cox & Snell) | Model LR (p) | Hosmer Lemeshow Test (p) | ß | Exp (ß) (CI 95%) |
|---|---|---|---|---|---|
| **SF-36—Domains** | | | | | |
| Physical role functioning | 0.124 | <0.001 | 13.359 (0.064) | 0.022 | 1.01 (1.01–1.03) |
| Physical functioning | 0.051 | 0.020 | 2.743 (0.254) | 0.010 | 1.01 (1.01–1.01) |
| **SF-36—Dimensions** | | | | | |
| Saúde física | 0.041 | 0.037 | 1.837 (0.055) | 0.029 | 1.02 (1.00–1.05) |
| **CCCUQ—Divisions** | | | | | |
| Social interaction | 0.157 | <0.001 | 13.180 (0.106) | -0.029 | 0.97 (0.95–0.98) |
| Domestic activities | 0.112 | <0.001 | 11.525 (0.174) | -0.022 | 0.97 (0.95–0.98) |
| Cosmesis | 0.083 | 0.003 | 13.143 (0.107) | -0.019 | 0.98 (0.96–0.99) |
| Emotional status | 0.047 | 0.026 | 12.616 (0.126) | -0.015 | 0.98 (0.96–0.99) |
| Total Score | 0.117 | <0.001 | 8.274 (0.407) | -0.027 | 0.97 (0.95–0.98) |

pronounced in the HG. In contrast, the UG was the only group to exhibit significant deteriorations, particularly in physical, emotional, and cosmesis states. These findings contribute valuable insights and update the body of knowledge on VLU treatment, reinforcing the concept that the effects of such treatments extend beyond mere lesion healing. This emphasizes the importance of considering the patient's entire context, including access to health resources and their sociodemographic environment [18].

The sociodemographic profile of our sample highlighted that the majority of VLU patients were women over 60 years of age, with low levels of education, and living in impoverished conditions. This aligns with existing literature that identifies older age, female gender, low income, and limited education as risk factors for VLU, due to their collective impact on venous circulation diseases [1, 8, 19–21]. Aging and being female increase hydrostatic pressure in the venous system, which intensifies over the years. Without intervention, the severity of venous insufficiency increases, leading to VLU formation, often compounded by health comorbidities [22]. Beyond age and gender, our data on income and education also warrant attention. A study conducted in Colombia found that worse prognoses for VLU outcomes correlated with poor social and economic situations [23]. Authors explain that socioeconomic status hinders access to more advanced treatment technologies, limiting patient options. Lower education levels may also result in less knowledge about the disease, reducing self-care, often exacerbated by the presence of comorbidities [3, 24]

The study also examined health aspects, noting the prevalence of Systemic Arterial Hypertension and Diabetes Mellitus in the sample, conditions known to compromise vascular flow and worsen VLU outcomes [1, 20]. Some authors suggest that changes in diet, improved sleep, and rest may assist in skin regeneration and anti-aging processes [25]. Although both groups were homogeneous in terms of sociodemographic and health profiles, their influence on participants' QoL outcomes cannot be discounted.

In the descriptive analysis of QoL, the UG exhibited better QoL scores than HG at T1, particularly in the Physical role functioning domain, Physical health Dimension (SF-36), and Social interaction (CCVUQ). These results placed the UG in a better condition at the start of the research. Analyzing the total sample (both groups together), improvements were noticed in most SF-36 domains, although no significant changes were observed between T1 and T2 in the CCVUQ variables. Comparative analyses between UG and HG revealed that the HG showed significant improvements in the Physical Role Functioning, Physical Functioning domains, and Physical Health Dimension (SF-36), as well as in all CCVUQ divisions: Cosmesis, Emotional Status, Social Interaction, and Domestic Activities. Conversely, the UG exhibited a decline in these same aspects. Despite the Sign test analysis showing significant improvements in some SF-36 domains in the UG, other aspects, including CCVUQ, showed a decline in QoL, which was not observed in the HG. Improvements in social functioning, physical functionality, and domestic activities (CCVUQ) stood out in the HG. Logistic regression analysis in the HG identified that the most impacted domains were Physical Role Functioning (SF-36) and Social Interaction and Domestic Activities (CCVUQ).Many of these aspects related to how individuals perceive their ability to perform daily tasks, such as walking, climbing stairs, lifting heavy objects, and engaging in basic activities like preparing food, dressing independently, and taking medication without assistance [5, 13]. Other studies have emphasized the importance of these functions for individuals with VLU, as pain and other implications of the lesion-related impairments lead to disability and increased dependence for certain tasks [9, 10]. This can foster feelings of uselessness and lead to depressive episodes and other mental health issues [26].

Another notable aspect of our results was the improvement in Social interaction and Cosmesis in the HG, as evidenced by the CCVUQ measurement. In the context of VLU, these

constructs are often related, as the lesion's appearance can cause embarrassment, leading to social withdrawal from activities such as neighborhood gatherings, social events, and celebrations. VLU patients often fear exposing their wounds, even when covered by dressings, and are concerned about the risk of trauma to the lesion, which could hinder treatment [27]. Thus, the improvement in social interaction in our sample may be explained by the enhancement in cosmesis, among other factors.

It is important to emphasize that the outcomes observed in both study groups may explain the improved QoL in the HG. However, considering the limited access to advanced VLU treatment technologies in the study setting, and the sociodemographic and health profiles of the participants, the one-year interval may not be sufficient to fully evaluate the treatment effectiveness. A retrospective study showed that the percentage of VLU healed after one year of treatment is approximately 50.0% to 60.0% [19], which aligns with our findings. Other specific factors were not explored in our study, such as the anatomical location of VLU and clinical characteristics like wound depth, appearance, and border, all of which are related to the complexity and prognosis of each case [22].

It is well-known that compression therapy is the gold standard for VLU treatment [28], and it should be combined with pharmacological and behavioral adjuvants [29]. However, the context of this study was the Brazilian public health system, which is often limited in terms of funding and public health policy implementation [30]. Furthermore, VLU treatment requires investment in human and material resources. A 2020 study using the Secure Anonymised Information Linkage Databank estimated that the direct costs for VLU treatment exceeded £2 billion per year, with the most expensive resource being district nurse visits [3].

### Limitation of study

One limitation of the study was the number of patients lost during the research, resulting in a smaller sample than initially anticipated. Additionally, some patients were already receiving treatment before the research began, while others were still being included, potentially giving some an advantage. Interviews were conducted during consultations, which may have caused discomfort or evoked memories and feelings that led to overestimated responses. To mitigate these biases, the data collection environment was structured to provide privacy and a comfortable, welcoming atmosphere. We also analyzed potential confounding factors that could interfere with our findings. As this is an observational study, the data do not necessarily imply causation, and the potential for generalizing the results is limited.

### Conclusion

Our findings indicate that QoL significantly improved in aspects related to functionality, physical performance, and social interaction in individuals whose lesions healed after one year of Unna boot compression therapy. Furthermore, the deterioration of these same aspects in the group with unhealed VLUs supports our study hypothesis.

These conclusions must be viewed within the context of the complex circumstances surrounding these patients, where emotional consequences may stem from a multidimensional reality that cannot be easily addressed by reducing lesion size alone. Therefore, further in-depth research is recommended to explore other potential factors that can be improved with the successful treatment of VLUs.

### Acknowledgments

We extend our sincere thanks to the patients who agreed to participate in our study and to the managers of the chronic wound treatment clinic, who supported us in the recruitment and

sample selection process. We are also grateful for their assistance in providing access to the clinic's physical space and the information available through the service.

## Author Contributions

**Conceptualization:** Mário Lins Galvão de Oliveira, Felipe León-Morillas, Isadora Costa Andriola.

**Data curation:** Gilson de Vasconcelos Torres.

**Formal analysis:** Gilson de Vasconcelos Torres.

**Funding acquisition:** Gilson de Vasconcelos Torres.

**Investigation:** Mário Lins Galvão de Oliveira, Isadora Costa Andriola.

**Methodology:** Mário Lins Galvão de Oliveira, Isadora Costa Andriola.

**Project administration:** Carmelo Sergio Gómez Martínez, Gilson de Vasconcelos Torres.

**Resources:** Mário Lins Galvão de Oliveira, Bruno Araújo da Silva Dantas.

**Supervision:** Felipe León-Morillas, Carmelo Sergio Gómez Martínez, Bruno Araújo da Silva Dantas.

**Validation:** Felipe León-Morillas, Bruno Araújo da Silva Dantas.

**Writing – original draft:** Mário Lins Galvão de Oliveira.

**Writing – review & editing:** Felipe León-Morillas, Carmelo Sergio Gómez Martínez, Bruno Araújo da Silva Dantas.

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
