## [Decision Letter · Decision Letter 0]

2 Oct 2024

PONE-D-24-08769Venous leg ulcer healing as a determinant of quality of life in patients treated with unna boot: a quasi-experimental studyPLOS ONE

Dear Dr. Dantas,

Thank you for submitting your manuscript to PLOS ONE. After careful consideration, we feel that it has merit but does not fully meet PLOS ONE’s publication criteria as it currently stands. Therefore, we invite you to submit a revised version of the manuscript that addresses the points raised during the review process.

 Please consider the additional information required as per the reviewers' comments below, to allow a full clinical picture of your participants for the reader.  As your inclusion criteria re ulcer duration indicates you only included chronic venous leg ulcers, be sure to specify this when discussing your sample.  A review of grammar and language is needed.

We look forward to receiving your revised manuscript.

Kind regards,

Kathleen Finlayson

Academic Editor

PLOS ONE

Journal Requirements:

 This study was funded by the National Council for Scientific and Technological Development (Brazil) through the CNPq/MCTI/FNDCT call, grant number 408535/2021-0 and document nº 18/2021 - Tier B, for consolidated groups. The grant was awarded to researcher Dr. Gilson de Vasconcelos Torres, from the Federal University of Rio Grande do Norte, Brazil (Level PQ1D Researcher).

Reviewers' comments:

Reviewer's Responses to Questions

**Comments to the Author**

1. Is the manuscript technically sound, and do the data support the conclusions?

Reviewer #1: Yes

Reviewer #2: Yes

2. Has the statistical analysis been performed appropriately and rigorously? 

Reviewer #1: Yes

Reviewer #2: Yes

3. Have the authors made all data underlying the findings in their manuscript fully available?

Reviewer #1: Yes

Reviewer #2: No

4. Is the manuscript presented in an intelligible fashion and written in standard English?

Reviewer #1: No

Reviewer #2: No

5. Review Comments to the Author

Reviewer #1: Dear Authors

Thank you for the opportunity to read the research results. I suggest making the following corrections/additions:

- in Introduction - population 1% - outdated data ...

- during treatment - how was the progress of ulcer healing assessed? what tool...?

- inclusions criteria - only age? nothing else?, gender, other data?

- what were the recommendations for patients undergoing compression therapy treatment (we have 1 year of observation here)

- what treatment was provided apart from Unna boot? (pharmacology, dressings?()

- I suggest English Native Speaker correction

Reviewer #2: Dear Author,

The manuscript is technically sound piece of scientific research with data that supports the conclusions. Clinical study has been conducted rigorously, with appropriate controls, replication, and sample sizes. statistical analysis is adequate and optimum. Data is presented well. English and grammar needs improvement.

Thanks

With Best Regards

6. PLOS authors have the option to publish the peer review history of their article (what does this mean?). If published, this will include your full peer review and any attached files.

Reviewer #1: No

Reviewer #2: No

---

## [Author Response · Author response to Decision Letter 0]

8 Oct 2024

Dear Reviewers,

On behalf of the authors, I would like to extend my greetings and express our gratitude for the thoughtful comments and requests regarding the submitted manuscript. I am confident that they have helped improve the quality of the work to which we have been so dedicated.

Below, I have provided point-by-point responses to each of your comments. I hope these answers satisfactorily address the concerns you raised.

Comments to the Author

1. Is the manuscript technically sound, and do the data support the conclusions?

Reviewer #1: Yes

Reviewer #2: Yes

2. Has the statistical analysis been performed appropriately and rigorously? 

Reviewer #1: Yes

Reviewer #2: Yes

3. Have the authors made all data underlying the findings in their manuscript fully available?

Reviewer #1: Yes

Reviewer #2: No

4. Is the manuscript presented in an intelligible fashion and written in standard English?

Reviewer #1: No

Reviewer #2: No

5. Review Comments to the Author

Reviewer #1: 

Dear Authors

Thank you for the opportunity to read the research results. I suggest making the following corrections/additions:

- in Introduction - population 1% - outdated data ...

Response: We found a meta-analysis published in 2023 that reports a prevalence of 0.32% and an incidence of 0.17%. We have replaced the theoretical reference accordingly. The study is described as follows:

Probst, S., Saini, C., Gschwind, G., Stefanelli, A., Bobbink, P., Pugliese, M. T., Cekic, S., Pastor, D., & Gethin, G. (2023). Prevalence and incidence of venous leg ulcers—A systematic review and meta-analysis. International Wound Journal, 20(9), 3906–3921. https://doi.org/10.1111/iwj.14272

- during treatment - how was the progress of ulcer healing assessed? what tool...?

Response: At each dressing change appointment at the service, a new evaluation was conducted by the healthcare professional. The evaluation focused on the same aspects, including the size and width of the wound, the appearance of its edges and interior, as well as the presence and characteristics of exudate. We have added this information to the methodology to make it clearer.

- inclusions criteria - only age? nothing else?, gender, other data?

Response: In addition to age as a requirement, the participant had to be registered with the primary care service in the study setting and not with other services. Furthermore, the venous ulcer had to be considered active, meaning it should not have healed. Even when covered by thin layers of skin, a wound could still be confused with an active venous ulcer. We have added a clarification in the methods section to better explain this point.

- what were the recommendations for patients undergoing compression therapy treatment (we have 1 year of observation here)

Response: Patients were routinely instructed on how to care for their dressings at home, such as protecting them during bathing to avoid getting them wet, following a healthy diet, and elevating the leg above heart level. All these instructions are in line with the conduct manual adopted as a theoretical reference cited in the manuscript. We have added a description of these instructions in the methodology to make it clearer.

- what treatment was provided apart from Unna boot? (pharmacology, dressings?()

Response: The treatment primarily consisted of an Unna boot covered by a secondary dressing (over the boot). Only in cases suggestive of wound infection did the physician implement antibiotic treatment before starting compression therapy. This is because infectious processes contraindicate the use of compression therapy. We have described this in the methodology.

- I suggest English Native Speaker correction

Response: We conducted a thorough revision of the English language with the assistance of a native speaker. The revision was both grammatical, and we aimed to make the language more fluid for better comprehension.

Reviewer #2: 

Dear Author,

The manuscript is technically sound piece of scientific research with data that supports the conclusions. Clinical study has been conducted rigorously, with appropriate controls, replication, and sample sizes. statistical analysis is adequate and optimum. Data is presented well. English and grammar needs improvement.

Thanks

With Best Regards

Response: Thank you for your comments and observations. We have conducted a thorough revision of the English language with the help of a native speaker. The revision focused on grammar, and we have worked to improve the fluidity of the language for better understanding.

---

## [Editor Report · Decision Letter 1]

12 Nov 2024

Venous leg ulcer healing as a determinant of quality of life in patients treated with unna boot: a quasi-experimental study

PONE-D-24-08769R1

Dear Dr. Dantas,

We’re pleased to inform you that your manuscript has been judged scientifically suitable for publication and will be formally accepted for publication once it meets all outstanding technical requirements.

Kind regards,

Kathleen Finlayson

Academic Editor

PLOS ONE

Additional Editor Comments:

Thank you for addressing the reviewers' suggestions, the manuscript now is clearer and provides further details on your methods.

There are a couple of remaining areas requiring some additional detail or clarification as follows:

- firstly, how was the status of 'healed' ulcer defined?

- in Table 5 (logistic regression), could you specify the outcome variable here? was it healed or unhealed? if so, I would change the title of the table description to reflect that the regression was on all participants, looking at factors associated with healing (rather than only including those who healed?

---

## [Editor Report · Acceptance letter]

14 Nov 2024

PONE-D-24-08769R1 

PLOS ONE

Dear Dr. Dantas, 

I'm pleased to inform you that your manuscript has been deemed suitable for publication in PLOS ONE. Congratulations! Your manuscript is now being handed over to our production team.

Kind regards, 

on behalf of

Dr. Kathleen Finlayson 

Academic Editor

PLOS ONE